# Debonding Damage Detection in CFRP Plate-Strengthened Steel Beam Using Electromechanical Impedance Technique

**DOI:** 10.3390/s19102296

**Published:** 2019-05-18

**Authors:** Bin Wang, Yongfeng Sun, Yunzhang Li, Chuan Zhang

**Affiliations:** 1School of Civil and Architecture Engineering, Xi’an Technological University, Xi’an 710021, China; sunyongfeng@st.xatu.edu.cn; 2School of Management, Xi’an University of Finance and Economics, Xi’an 710100, China; yunzhang_0808@xaufe.edu.cn; 3Key Laboratory of Transportation Tunnel Engineering, Ministry of Education, Southwest Jiaotong University, Chengdu 610031, China

**Keywords:** carbon fiber reinforced polymer, lead zirconate titanate transducer, debonding detection, electromechanical impedance, cross-correlation coefficient

## Abstract

Carbon fiber reinforced polymer materials are widely applied in structure strengthened engineering because of the many advantages of carbon fiber reinforced polymer. However, the debonding damage between the carbon fiber reinforced polymer and host structures occurs frequently, which might lead to the brittle failure of structure components, especially flexural ones. In this paper, an electromechanical impedance-based method, an important technique in structural health monitoring, was adopted to detect the debonding damage of carbon fiber reinforced polymer plate-strengthened steel beam by using lead zirconate titanate (PZT) transducers. A carbon fiber reinforced polymer plate-strengthened steel beam specimen was fabricated in the laboratory and two PZT sensors were attached at different locations on the carbon fiber reinforced polymer plate. The impedance signatures with variation of the different degrees of the debonding damage were measured by an impedance analyzer. The root-mean-square deviation method and the cross-correlation coefficient method were used to quantify the correlation between the electromechanical impedance and the debonding damage degree. The results reflect that an electromechanical impedance-based structural health monitoring technique can serve as a good method to detect the debonding damage of carbon fiber reinforced polymer plate-strengthened steel structures.

## 1. Introduction

Concrete and steel are the most common construction materials, however, they are subject to corrosion [1,2], fatigue [3,4], and other adverse effects, which often lead to structural damage and may reduce the structural bearing capability and durability if not repaired in time. Therefore, to ensure the safety and normal function of these structures, it is necessary to conduct a reliability evaluation of and reinforcement to these structural damages. With strong corrosion resistance, fiber reinforced polymer (FRP) has been widely used in civil engineering [5,6], including strengthening civil structures, especially the bending components [2,3,4]. Although at a higher cost, carbon fiber reinforced polymer (CFRP), because of its advantages of high tensile strength, high elastic modulus, good durability, and light weight, is being increasingly used in structural repair and strengthening [1,5].

The existing research shows that the bond between CFRP and the structure is usually the weakest link, and the debonding of the CFRP from the structure substrate is one of the main failure modes, which can cause the brittle and sudden failure of structures [7,8,9]. Meanwhile, some experimental studies have also shown that the CFRP plates possess more advantages than the CFRP sheets for strengthening the damaged flexural components [10,11]. However, the CFRP plates are more likely to generate debonding damage than the CFRP sheets, especially at the two ends of the components under loading. [12]. At present, there are many studies on the debonding damage of CFRP plate-strengthened reinforced concrete structure components [13,14,15], but very few studies on CFRP plate-strengthened steel structure components have been reported, especially for CFRP plate-strengthened bending steel components [16]. Therefore, this paper conducts an experimental research on the debonding damage of the CFRP plate-strengthened steel structure beam.

Different to the failure form of CFRP sheet-strengthened concrete structures, in the CFRP plate-strengthened steel structures, the steel and CFRP plates possess much higher failure strength than the adhesive bond layer [17,18]. As shown in Figure 1, the debonding failure types of the CFRP plate-strengthened steel structures are given as follows and mainly include: (a) adhesion failure at the CFRP/adhesive interface; (b) adhesion failure at the steel/adhesive interface; (c) cohesive failure in the adhesive layer and (d) CFRP plate delamination. The main reason for the debonding damage is that the adhesive layer is subject to overload. Meanwhile, under the bending effect of the beam component, the bending deformation of the CFRP plate presents an inconsistent characteristic, which is also an important cause of the debonding damage. On the other hand, due to the material discontinuity at the location of the bond end, the stress concentration in these areas may cause the failure of the adhesive before the steel and the CFRP plate both reach their ultimate strengths. Such a steel beam may fail due to the debonding of the CFRP plate that generally initiates at one of the plate ends. This failure mode not only wastes the high performance of the CFRP, but also greatly reduces the reliability of the strengthened components [19].

Since debonding damage is a type of brittle failure, therefore, an effective non-destructive monitor technique to monitor the initial debonding failure becomes essential in this application. This can help provide timely references to take further measures and avoid serious failures. [20]. Non-destructive monitoring techniques, such as acoustic inspection technology [21,22], ultrasonic inspection technology [23,24,25], infrared thermography [26,27], fiber optic sensing [28,29,30,31] and X-ray inspection [32,33] have been used extensively to detect structure damage. However, these conventional inspection technologies require complex algorithms and equipment, which is more difficult for real engineering applications.

In recent years, lead zirconate titanate (PZT) materials, which have many advantages such as small size, light weight, low cost, good stability, etc., have been widely applied in structural health monitoring (SHM) [34,35,36,37,38,39]. Based on the PZT materials, a wide range of methods have been developed, including the smart aggregate enabled active sensing method [40,41,42,43,44], the electromechanical impedance (EMI)-based damage monitoring technique [45,46,47,48,49] and some other methods [50,51,52,53,54,55]. As a new technology for the non-destructive testing (NDT), the EMI technique largely attracts researchers’ attention, and EMI methods have many advantages such as sensitivity to the initial damage of the structure, strong anti-interference ability in the environment, and the dual effects of sensing and driving. Thus, compared with active sensing technology, the number of PZT patches can be reduced, and without relying on model analysis. The EMI technique has been successfully applied to steel and concrete crack detection [56,57,58,59], concrete strength monitoring [60,61] and corrosion monitoring of reinforced concrete [62] among others. In recent years, the EMI technique has been used in the debonding monitoring on CFRP strengthened concrete structures [63,64,65,66]. However, to the authors’ knowledge, few studies have been carried out on the use of the impedance measurement on the debonding of CFRP plate-strengthened steel structures.

In the present study, the different debonding damage states were investigated by analyzing the electromechanical impedance variations of the piezoelectric sensors. The root-mean-square deviation (RMSD) method and cross-correlation coefficient (CC) method were used to quantify the correlation between the impedance signatures and the debonding states. Finally, the experimental results demonstrate that the debonding damage of the CFRP plate-strengthened steel beam can be effectively monitored by adopting the EMI-based method.

## 2. EMI-Based SHM Technique

### 2.1. Technical Background

The EMI method is used to monitor the debonding damage of the CFRP plate-strengthened steel beam by the PZT-based transducers in this paper. Due to the fact that the PZT transducer has direct and inverse piezoelectric effects, the EMI method can employ one PZT patch both as an actuator and a sensor at the same time. The electrical impedance of the PZT bonded onto a host structure is directly related to the mechanical impedance of the structure. The interaction between the PZT patch and the host structure can be idealized as an electromechanical coupling system. Thus, the change in the mechanical impedance of the structure caused by damage, which leads to the reduction of stiffness and/or damping of structure, is reflected by the change in the electrical impedance of the PZT patch. Thus, the presence of damage and its severity to the host structure can be detected by analyzing the electrical impedance signatures of the PZT patch [67]. The one-dimensional, electromechanical system formed by the interaction between the PZT patch and a host structure was first established by Liang et al. [68] and was then studied by many other researchers [56,57,58,59,60,61,62,63,64,65,66]. An illustration is shown in Figure 2.

Since the PZT patch has the deformation in the x-direction and y-direction, the one-dimensional electromechanical model only considers the deformation in one direction, and the deformation in the other direction is ignored. Therefore, the two-dimensional electromechanical model can more accurately reflect the effect of electromechanical coupling between the PZT patch and the host structure [69,70]. The one-dimensional and two-dimensional electromechanical models ignore the interaction effect of the longitudinal vibration of the PZT patch for the vibration of the embedded PZT patch. The three-dimensional electromechanical model was proposed [71,72]. However, the two-dimensional and three-dimensional electromechanical models are complex and cannot reflect the influence of PZT mechanical impedance and structural mechanical impedance on the coupled conductance. It is still difficult to apply in engineering practices. The one-dimensional, electromechanical model has many advantages, including clear physical meaning, simple model, and good one-to-one correspondence between parameters, which facilitates structural parameter identification and health monitoring. Therefore, in this paper, a one-dimensional model is used and expressed as [68]
(1)$$Y\left(\omega \right)=G\left(\omega \right)+B\left(\omega \right)j=\omega j\frac{wl}{h}\left[{\bar{\varepsilon }}_{33}^{T}-d_{31}^{2}{\bar{Y}}^{E}+d_{31}^{2}{\bar{Y}}^{E}\left(\frac{Z_{a}\left(\omega \right)}{Z_{s}\left(\omega \right)+Z_{a}\left(\omega \right)}\right)\left(\frac{\tan\kappa l}{\kappa l}\right)\right]$$
where $d_{31}^{2}$ is the piezoelectric strain coefficient of the PZT material, ${\bar{Y}}^{E}$ is the complex Young’ s modulus under constant electric field, κ is the wave number, ν is Poisson’s ratio, G is the conductance, B is the susceptance, j is the imaginary unit, $Z_{s}\left(\omega \right)$ and $Z_{a}\left(\omega \right)$ are the electrical impedance of a PZT patch and host structure, respectively, ${\bar{\varepsilon }}_{33}^{T}$ is the complex electric permittivity at constant stress, w, l and h are the width, length and thickness of a PZT patch, respectively.

As indicated in Equation (1), once the CFRP plate-strengthened steel beam experiences the debonding damage, the mechanical impedance of the structure will change, which can be further reflected by the coupled electrical impedance of the PZT patch. This serves as the principle for monitoring the debonding damage of the CFRP plate-strengthened steel beam through the EMI method.

### 2.2. Statistical Damage Indices: Root-Mean-Square Deviation (RMSD) and Cross-Correlation Coefficient

In order to assess the degree of the debonding damage, in this paper, the root-mean-square deviation (RMSD) is adopted to quantify the overall change of the impedance signature with variation of the debonding damage. The mathematical expression of the RMSD is written as [73]
(2)RSMD(%)=∑i=1n{Re(Zi,1)−Re(Zi,0)}2∑i=1n{Re(Zi,0)}2×100%
where Re denotes the real part of the electromechanical impedance, $Z_{i,0}$ is the impedance of the PZT measured in the initial state (perfect bonding), $Z_{i,1}$ is the impedance of the PZT measured in the concurrent debonding damage state and n is the number of the frequency points.

With the RMSD index, a greater numerical value of the index indicates a larger difference between the baseline reading and the subsequent reading, which further reflects the degree of the dynamic debonding damage. Moreover, another statistic index, named as the cross-correlation coefficient (CC), is also used as an index to characterize the debonding damage of the CFRP plate-strengthened steel beam and its mathematical expression [74] can be written as
(3)$$\mathrm{CC}=\frac{\frac{1}{N}\sum_{i=1}^{N}\left(x_{i}-\bar{x}\right)\left(y_{i}-\bar{y}\right)}{\sigma_{x}\sigma_{y}}$$
where $\bar{x}$ and $\bar{y}$ are the mean values of the two data sets, respectively, and $\sigma_{x}$ and $\sigma_{y}$ are the standard deviations of the signature data x and y, respectively. The Equation (3) shows that the more closely correlated the two signatures are, the closer the CC is to 1. With an increase to the degree of debonding damage, the damage index increases, therefore, normally using “1-CC” instead of CC. 

At present, the above statistical damage index methods have been applied in the analysis of the damage monitoring and strength development of concrete [58,60], debonding damage monitoring of FRP strengthened RC Beams [62], debonding damage monitoring of CFRP sheet-strengthened concrete structures and fiber-reinforced polymer rebar–reinforced concrete [63,66], and concrete grouting density monitoring of concrete-filled fiber-reinforced polymer tube [64]. However, these damage-index-based methods have not been applied in the analysis of the debonding damage of CFRP plate-strengthened steel structural members based on the EMI method. Therefore, in this paper, the RSMD and CC statistical damage indices will be adopted to analyze the debonding damage of CFRP plate-strengthened steel beams.

## 3. Experimental Details

### 3.1. Preparation of Test Specimen

In this paper, a CFRP plate-strengthened steel beam specimen was prepared for debonding damage monitoring based on the EMI method. The lengths of the I-shaped steel beam specimen and the CFRP plate were 1.3 m and 1.1 m, respectively. The CFRP plate was bonded on the steel beam with epoxy glue. Two PZT sensors (PZT-1 and PZT-2), which had a size of 10 mm × 10 mm × 1 mm, were bonded on the surface of the CFRP plate. The PZT-2 sensor was located in the midspan of the CFRP plate, while the PZT-1 sensor was located 15 cm away from the left end of CFRP plate. The geometry of the CFRP plate-strengthened steel beam specimen along with the layout of the PZT sensors are shown in Figure 3.

The debonding damage between the CFRP plate and the steel beam was produced by using tools such as a grinder and scalpel. Due to the brittleness performance of debonding failure, it is difficult to obtain the predesigned debonding length in this test. Therefore, the five different debonding lengths, namely, 0.0 cm (D0, perfect bonding), 3.0 cm (D1), 6.5 cm (D2), 10.5 cm (D3) and 13.5 cm (D4), were obtained in sequence at the right side of the specimen and the real-time impedance measurements were performed at each debonding damage state, as shown in Figure 4.

In order to further verify the sensitivity of the PZT patch to the debonding damage distance, in this test, when the debonding length reached the PZT-1 position, the debonding length was increased continually, and then the PZT-2 patch was used to monitor the debonding damage. The distance from debonding damage position to PZT-2 patch was 32–33 cm (D5), 15–16 cm (D6), 12–13 cm (D7) and 10–11 cm (D8), respectively.

### 3.2. EMI Monitoring

The EMI testing and the data acquisition system consisted of two PZT patches, one impedance analyzer, and one personal computer equipped with the designed data acquisition software, as shown in Figure 5. The impedance signatures of the PZT patches at the aforementioned five debonding damage states were recorded by an Agilent 4294 A impedance analyzer. The impedance spectra, including the real and imaginary parts, namely, the resistance and the reactance of the PZT patches, were measured for further analysis. It should be mentioned that the resistance had already been verified to be a more sensitive and stable indicator than the reactance, thus only the statistical analysis of the resistance was performed in this study [58,60].

The EMI spectra of the bonded PZT transducer were measured by sweeping a frequency range from 50 kHz to 400 kHz in this test. Figure 6 shows the real part of impedance signatures within the frequency band of 50 kHz–400 kHz, which were measured by the PZT-1 and PZT-2 transducers at the non-damage state of the CFRP plate-strengthened steel beam. It can be seen that the two signatures coincide basically both in the shape and magnitude of the spectra. There are two observed resonant frequency peaks (RFP) approximately at the frequency range from 140 kHz to 260 kHz, as shown in Figure 6. In this paper, the second RFP with a larger magnitude at the frequency range from 200 to 260 kHz was focused on for further data analysis.

## 4. Results and Discussion

Figure 7 displays the real parts of the impedance signatures of the two different PZT-based impedance transducers at different states of debonding damage. As can be seen from Figure 7a, the resistance signature obtained by the PZT-1 presents more noticeable changes at different debonding damage states. The peak resistance decreases significantly with the increase in the length of debonding, especially at the initial stage of debonding damage, where the debonding damage state aggravates from D0 to D1. Meanwhile, the frequency corresponding to the peak resistance gradually moves to the lower frequency from D0 to D4. As compared with the resistance obtained by the PZT-1, the resistance obtained by the PZT-2 is basically unchanged from D0 to D4. This is mainly because the location of the debonding damage still has a certain distance from the area of debonding. Thus, the PZT-2 is less sensitive to the debonding damage compared with the PZT-1 transducer.

As mentioned above, to estimate the effective distance of the PZT transducer for debonding damage monitoring, in this test, the debonding length was increased continually when the debonding damage reached the PZT-1 position. At this time, PZT-2 was used to detect the debonding damage monitoring. The results are shown in Figure 7b. As can be seen, when the debonding damage reached D5 and D6, the peak resistance decreases slightly which is not significant. And the corresponding frequency did not change. When the debonding damage reaches D7, the peak resistance is significantly decreased, and the corresponding frequency gradually moves to the lower frequency. So based on the test results, in this paper the effective range of the PZT patch for debonding damage monitoring of the CFRP plate-strengthened steel beam is about 12 cm from the PZT patch. Hereafter, the debonding damage continues to increase and the monitoring results are similar to the results obtained by PZT-1. Hence, the real part of impedance signatures obtained by PZT-1 is used in the analysis in the next section.

The changes in the magnitude of the peak resistance and the frequency corresponding to the peak resistance with the development of the debonding damages are shown in Figure 8a,b, respectively. It can be seen from the figure that the resonant frequency and the resistance magnitude maintain similar trends under the different debonding damage states. The relative changes to the peak resistance decrease by 20.57%, 15.64%, 5.46%, 11.06% at the four different debonding damage states, respectively, while the relative changes in the frequency corresponding to the peak resistance under different debonding damage states decrease by 0.63%, 1.8%, 0.86%, 0.54%, respectively. As compared with the change in the resonant frequency, the change in peak resistance under different debonding damage states is more obvious. This is mainly due to significant changes in structure stiffness caused by the dynamic debonding damage, which results in significant changes to the mechanical impedances of the strengthened steel beam. This demonstrates that both the change in the resonant frequency and the change in the peak resistance value can be an effective indicator to monitor the debonding damage of the CFRP plate-strengthened steel structure.

As mentioned above, some statistic indices, such as the RMSD and the CC, can reflect the overall difference of impedance curves under different debonding damage states, thus making them an alternative method to determine the degree of the debonding damage. The RMSD of the resistance signatures of the PZT-1 transducer was obtained, as shown in Figure 9. It can be seen from the figure that the RMSD value increases gradually with the increase in the degree of damage. A larger RMSD value indicates a more severe debonding damage, especially at the initial stage of debonding damage, which confirms the PZT sensors are very sensitive in identifying change in the debonding damage. After that, the RMSD index still goes up but with a gradually decreasing rate.

According to Equation (3), the 1-CC values were also calculated to quantify the overall changes in the resistance signatures of the PZT-1 transducer attached to the specimen with different debonding lengths of 0 cm, 3 cm, 6.5 cm, 10.5 cm and 13.5 cm, respectively. Table 1 shows that as the CFRP debonding length increases from 0 to 13.5 cm, the 1-CC-based damage metric changes as follows: 0, 0.076, 0.26, 0.31 and 0.37 at PZT-1. Using 1-CC values of PZT-1, the 93.9% confidence level threshold of the intact condition was set using the generalized extreme value distribution. The relationship between the 1-CC and the different debonding damages is displayed in Figure 10. It shows that the 1-CC increases gradually with the increase in the degree of the debonding damage, which shares a very similar trend to that between the RMSD and the debonding damage. Based on the above experiment research, it can be concluded that both the RMSD and the CC can be adopted to quantitatively monitor the debonding damage, which further demonstrates that the EMI method is effective in detecting the debonding damage of the CFRP plate-strengthened steel structures.

## 5. Conclusions and Future Work

In this paper, the applicability of the electromechanical impedance-based method in monitoring the debonding damage of the carbon fiber reinforced polymer (CFRP) plate-strengthened steel structure was studied. A CFRP plate-strengthened steel beam was fabricated in the laboratory. Based on the method, two lead zirconate titanate (PZT) sensors were adopted to conduct the experiments. The results show that the real parts of the impedance signatures decrease with the increase in the debonding damage degree. In addition, the lead zirconate titanate sensor should be attached close to the debonding damage area, in order to ensure adequate sensitivity. The real parts of the impedance signatures of the lead zirconate titanate transducer show obvious changes when the debonding damage length changes. The statistical damage indices root-mean-square deviation and cross-correlation coefficient were both adopted to quantify the degree of the debonding damage. The values of root-mean-square deviation and cross-correlation coefficient increase obviously with the increase in the debonding length of carbon fiber reinforced polymer plate. The two damage indices clearly monitor the debonding damage evolution. The experimental results show that the electromechanical impedance method can be used to detect the debonding damage of the carbon fiber reinforced polymer plate-strengthened steel structures effectively.

Future work will involve testing CFRP strengthened steel specimens, which consider different design factors, with the PZT patch under loading. During the tests, the change of debonding length will be detected by the PZT patch under different loadings. We can use the sensor to provide real-time data on structural health monitoring for in-service structures with integrated sensors. Meanwhile, we will further conduct some investigation into the finite element analysis (FEA) method of CFRP-strengthened steel specimens. The correctness of the FEA method is further verified by comparing with the theoretical calculation results and the experimental results.

## Figures and Tables

**Figure 1 sensors-19-02296-f001:**
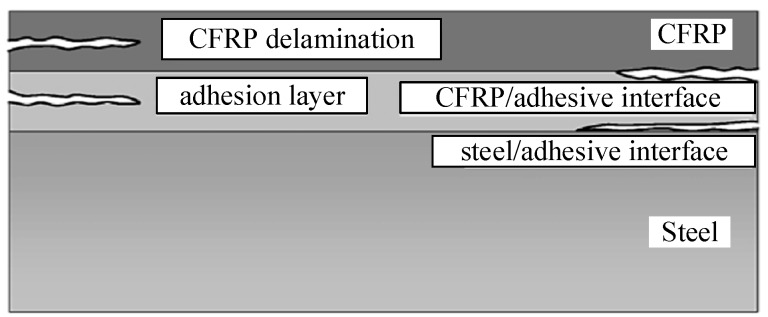
Form of the debonding failure for the carbon fiber reinforced polymer (CFRP) plate-strengthened steel components.

**Figure 2 sensors-19-02296-f002:**
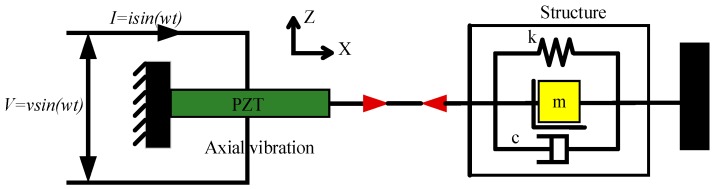
One-dimensional model of lead zirconate titanate (PZT)—structure dynamic interaction.

**Figure 3 sensors-19-02296-f003:**
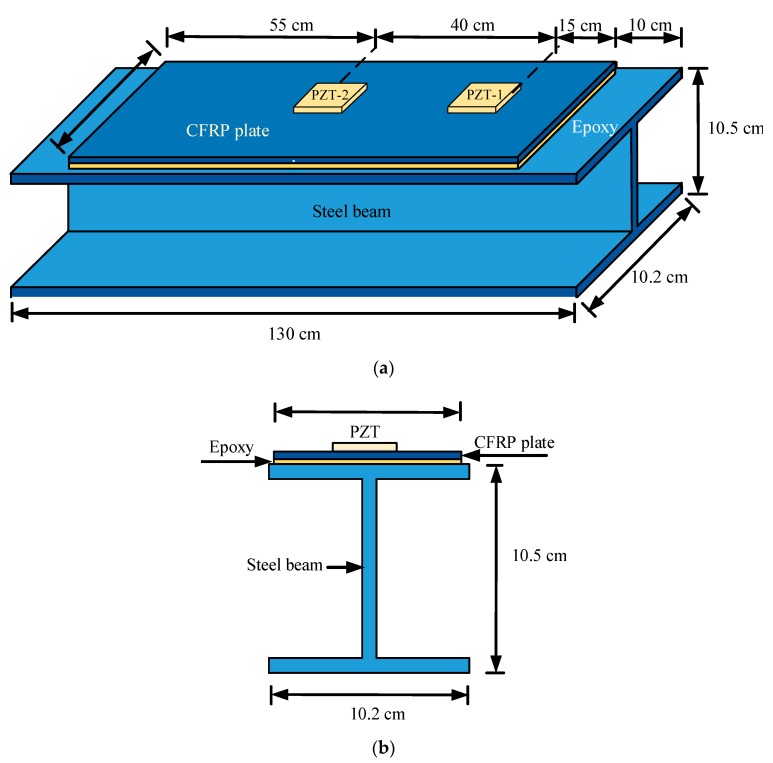
(**a**) CFRP plate strengthened steel beam specimen; (**b**) cross section of the CFRP plate strengthened steel beam.

**Figure 4 sensors-19-02296-f004:**
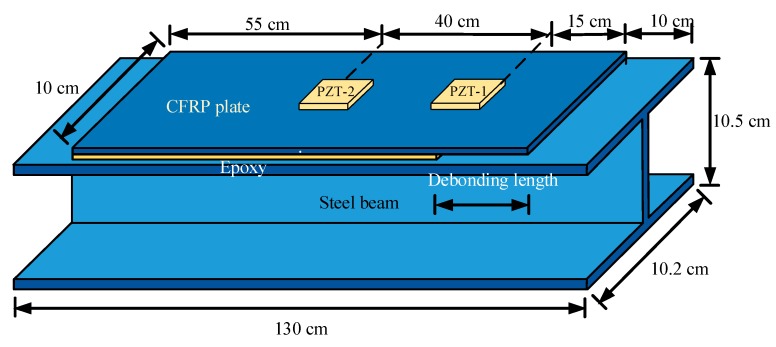
Specimen with CFRP plate debonding state.

**Figure 5 sensors-19-02296-f005:**
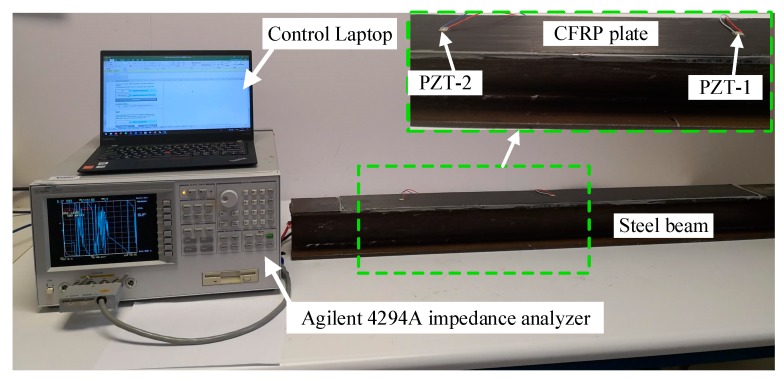
Experimental setup and PZT patches attached to the host structure.

**Figure 6 sensors-19-02296-f006:**
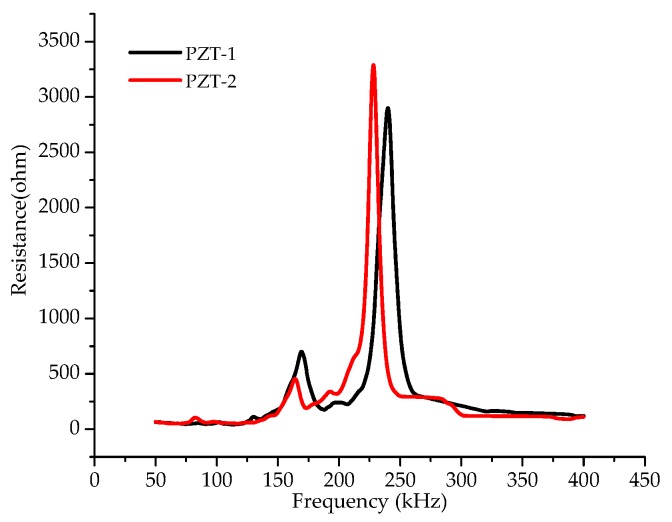
The resistance signatures at the frequency range from 50 kHz to 400 kHz of the PZT-1 and PZT-2 transducers at the perfect bonding state.

**Figure 7 sensors-19-02296-f007:**
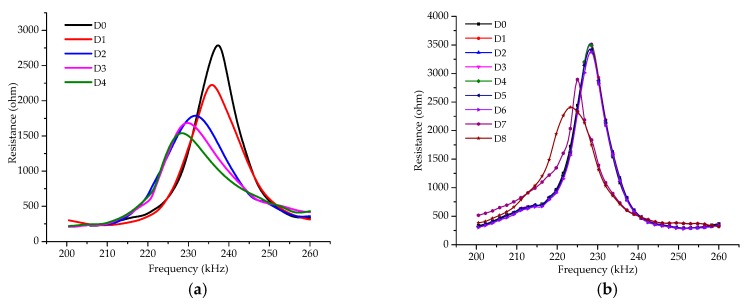
(**a**) Resistance spectra of the PZT-1; (**b**) resistance spectra of the PZT-2.

**Figure 8 sensors-19-02296-f008:**
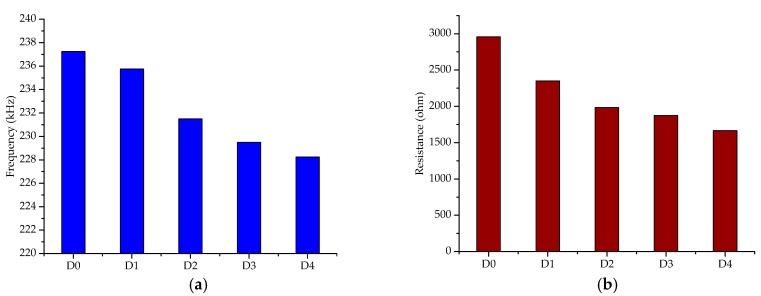
Changes in the peak resistance and frequency corresponding to the peak resistance with the different debonding damage states at PZT-1. (**a**) Changes of frequency corresponding to the peak resistance; (**b**) changes in the peak resistance.

**Figure 9 sensors-19-02296-f009:**
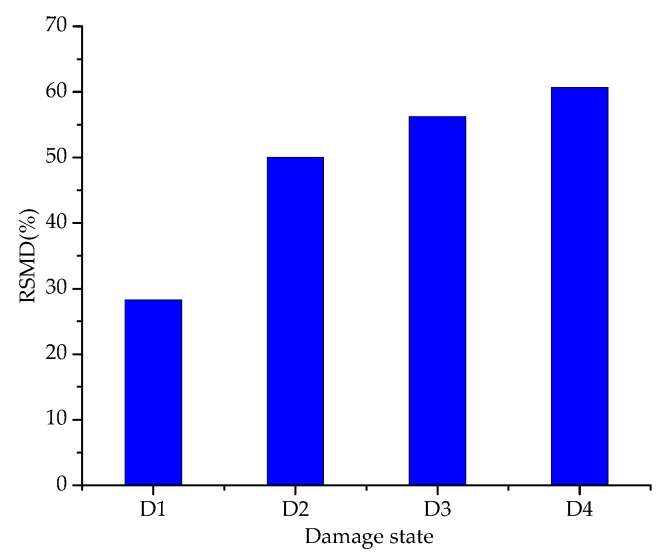
The value of root-mean-square-deviation (RMSD) for different debonding damage at the frequency range of 200–260 kHz at PZT-1.

**Figure 10 sensors-19-02296-f010:**
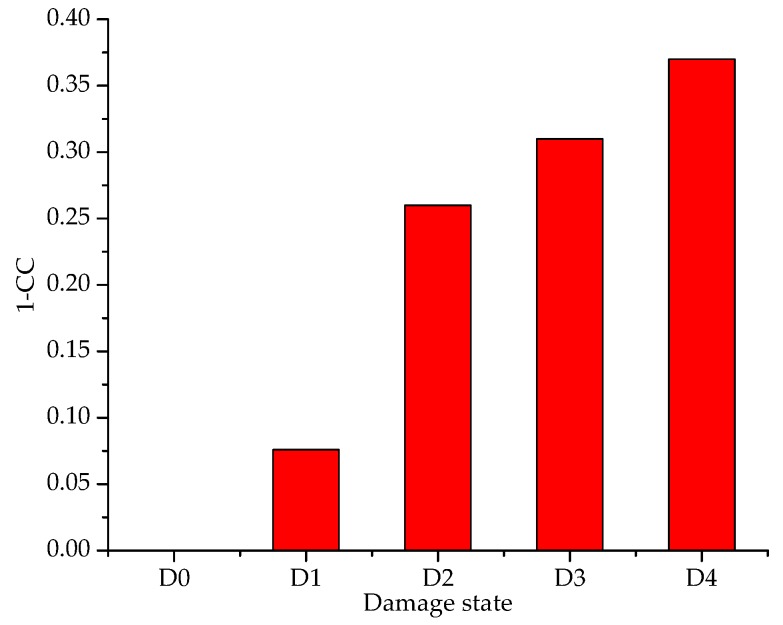
A 1-CC bar chart at PZT-1.

**Table 1 sensors-19-02296-t001:** Damage index: 1-CC at PZT-1.

Damage State	1-CC
D0	0
D1	0.076
D2	0.26
D3	0.31
D4	0.37

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
