# Peer review of "Debonding Damage Detection in CFRP Plate-Strengthened Steel Beam Using Electromechanical Impedance Technique"

_sensors, 2019, doi:10.3390/s19102296_

Round 1
Reviewer 1 Report
The manuscript describes the use of PZt based impedenace monitoring to detect debonding in CFRP plate strengthen steel structure. When the bonding layer was partially removed, the closest PZT experienced a measurable change in impedance. The paper shows an innovative application of EMI monitoring and just requires minor revisions:
1. There are quite a few misspellings and grammatical errors throughout the paper, and a minor proofreading is recommended. For example:
· Line 42 increasing to increasingly
· Line 56 – can bear -> possesses
· Line 71 – failures -> failure
· Line 114 – researcher – researchers
· Figure 2 – cibration – vibration
· Line 175 – scalper – scalpel
2. It seems PZT2 was not affected by the loss of bonding at the edge. Is there an estimated effective distance of the EMI monitoring technique (e.g. what’s the estimated number of PZTs per unit of area)? Can you estimate the effective range based on the data from PZT1?
3. Please comment on the advantages of EMI over other PZT based techniques, such as active sensing.
4. Since CFRP debonding may be due to bending and stress concentrations, would a strain gauge be good enough for detection of debonding?
Author Response
Response to Reviewer 1’s Comments
The authors greatly appreciate the time and effort by the reviewer for improving the quality of this manuscript. All the comments indicated in the reviewers’ report have been properly addressed and revised in the latest manuscript. The revisions are printed with Red Color.
1. There are quite a few misspellings and grammatical errors throughout the paper, and a minor proofreading is recommended. For example:
·Line 42 increasing to increasingly
·Line 56 – can bear -> possesses
·Line 71 – failures -> failure
·Line 114 – researcher – researchers
·Figure 2 – cibration – vibration
·Line 175 – scalper – scalpel
Response: The authors greatly appreciate the reviewer’s comment. We have proofread the manuscript carefully, and the writing has been improved. Some examples are given as follows,
Line 42: the “increasing” has been amended as “increasingly”.
Line 56: the “can bear” has been revised as “possesses”.
Line 71: the “failures” has been amended as “failure”.
Line 114: the “researcher” has been amended as “researchers”.
Figure 2: the “cibration” has been revised as “vibration”.
Line 175: the “scalper” has been amended as “scalpel”.
2. It seems PZT2 was not affected by the loss of bonding at the edge. Is there an estimated effective distance of the EMI monitoring technique (e.g. what’s the estimated number of PZTs per unit of area)? Can you estimate the effective range based on the data from PZT-1?
Response: The comment is considered. Actually, one of the purposes in this test is to estimate the effective distance of PZT patch to debonding damage, however the corresponding test results were not given in the manuscript, and now it is added in the revised manuscript.
3. Please comment on the advantages of EMI over other PZT based techniques, such as active sensing.
Response: The comment is considered. The author added this part of the introduction to the introduction.
4. Since CFRP debonding may be due to bending and stress concentrations, would a strain gauge be good enough for detection of debonding?
Response: The authors greatly appreciate the reviewer’s comment. In fact, strain gauges can detect debonding damage, and the relative research results have been reported. such as,
(1) Chiew, S.P.; Yu, Y.; Lee, C.K. Bond failure of steel beams strengthened with FRP laminates – Part 1: model development. Compos.: Part B, 2011, 42, 1114.
(2) Rahimi, H.; Hutchinson, A. Concrete beams strengthened with externally bonded FRP plates. J. Compos. Constr., 2001, 5, 44.
Reviewer 2 Report
The idea of the article is very average, "debonding"studies. This could have been a good paper if published 5 to 8 years ago. However, now it looks average. I suggest the authors improve the article as much as possible before acceptance.
Can you add numerical study to show how the contact/debonding effects, or else add some type of numerical model? for at least 2 cases of debonding to show the difference. or justify what parameters need to be adopted for numerical modelling, how to model de-bonding etc.
Susceptance also plays an important role in debonding even though it may not be considered for damages. include it to show its effect as I read susceptance play a significant role for debonding or delamination.
Paper is more like a report, need something more in the introduction, add any practical experience or failure where such a situation occurred in the past?
Have you tried different adhesives? any comment or influence of adhesives in your experiments.
I came across 3D and 2D EMI studies, why don't you include their studies in the intro (cite their work), and then mention that the present study can be idealised only to 1D study. At least readers will appreciate the fact of the existence of 1D, 2D and 3D models. Improve the introduction.
The scale of work is less, you need to make it more interesting by adding tables to show differences between damage indices.
willing to review again, all the best.
Author Response
Response to Reviewer 2’s Comments
The authors greatly appreciate the time and effort by the reviewer for improving the quality of this manuscript. All the comments indicated in the reviewers’ report have been properly addressed and revised in the latest manuscript. The revisions are printed with Red Color.
1. Can you add numerical study to show how the contact/debonding effects, or else add some type of numerical model? for at least 2 cases of debonding to show the difference. or justify what parameters need to be adopted for numerical modelling, how to model de-bonding etc.
Response: Thanks for the question and comments. The purpose of this work is to demonstrate the feasibility of using PZT enabled EMI approach to detect the debond between the FRP and the steel structure. In the future work, the author will carry out real-time monitoring of the debonding damage of CFRP strengthened beams under loading, and simulate the debonding damage by numerical analysis method.
2. Have you tried different adhesives? any comment or influence of adhesives in your experiments.
Response: The comment is considered. Since the purpose of this test is only to verify the effectiveness of EMI technology in monitoring CFRP plate strengthened steel beams, the analysis of factors, including different adhesives, which affect the debonding has not been carried out. The author will study health monitoring of the debonding damage under different influencing factors in further work.
3. I came across 3D and 2D EMI studies, why don't you include their studies in the intro (cite their work), and then mention that the present study can be idealised only to 1D study. At least readers will appreciate the fact of the existence of 1D, 2D and 3D models. Improve the introduction.
Response: The comment is considered. The authors have added the content about 3D and 2D EMI model in revised manuscript.
4. The scale of work is less, you need to make it more interesting by adding tables to show differences between damage indices.
Response: The comment is addressed in the revised paper. The authors have added a table in revised manuscript.
Reviewer 3 Report
Nice paper. I've some general comments and one major comment:
GENERAL COMMENT:
- Abstract, Keywords and Conclusion sections: please avoid using abbreviations and acronyms - these must be included in the main text only.
- Please, review carefully the English along the whole manuscript. It must be improved before submission, from the Abstract till Conclusions.
-Line 97-98-99: "Finally...." please move this to the Conclusion section and replace this with something more general, something like "Conclusion are drawn in the last section of the paper."
- Line 102: Please cancel "An important SHM technique" as there is no one technique more or less important than another one.
- Line 210- 211. Please avoid saying "right direction" but use something more scientific like "toward lower frequencies" or "upper frequencies" for example.
- Please include: "Laureti, S., Ricci, M., Mohamed, M. N. I. B., Senni, L., Davis, L. A. J., & Hutchins, D. A. (2018). Detection of rebars in concrete using advanced ultrasonic pulse compression techniques. Ultrasonics, 85, 31-38." in UT technique.
MAJOR COMMENT:
The Authors used only one transducer for obtaining the results. It would be very nice to see the spatial sensitivity limit of the transducer with respect to the various distance from the defect. In other words, at what distance (minimum) from the defect should the transducer be located to correctly recognise the defect?
Author Response
Response to Reviewer 3’s Comments
The authors greatly appreciate the time and effort by the reviewer for improving the quality of this manuscript. All the comments indicated in the reviewers’ report have been properly addressed and revised in the latest manuscript. The revisions are printed with Red Color.
1. GENERAL COMMENT:
- Abstract, Keywords and Conclusion sections: please avoid using abbreviations and acronyms - these must be included in the main text only.
- Please, review carefully the English along the whole manuscript. It must be improved before submission, from the Abstract till Conclusions.
-Line 97-98-99: "Finally...." please move this to the Conclusion section and replace this with something more general, something like "Conclusion are drawn in the last section of the paper."
- Line 102: Please cancel "An important SHM technique" as there is no one technique more or less important than another one.
- Line 210- 211. Please avoid saying "right direction" but use something more scientific like "toward lower frequencies" or "upper frequencies" for example.
- Please include: "Laureti, S., Ricci, M., Mohamed, M. N. I. B., Senni, L., Davis, L. A. J., & Hutchins, D. A. (2018). Detection of rebars in concrete using advanced ultrasonic pulse compression techniques. Ultrasonics, 85, 31-38." in UT technique.
Response: The authors greatly appreciate the reviewer’s comment. We have proofread the manuscript carefully, and the writing has been improved. Some examples are given as follows,
The abbreviations and acronyms in abstract, keywords and conclusion sections have been revised.
Line 97-98-99: "Finally...." has been moved to the Conclusion section and replaced them by simple sentence.
Line 102: "An important SHM technique" has been deleted.
Line 210- 211: "right direction" has been revised as "toward lower frequencies".
The "Laureti, S., Ricci, M., Mohamed, M. N. I. B., Senni, L., Davis, L. A. J., & Hutchins, D. A. (2018). Detection of rebars in concrete using advanced ultrasonic pulse compression techniques. Ultrasonics, 85, 31-38." has been cited in reference.
2. MAJOR COMMENT:
The Authors used only one transducer for obtaining the results. It would be very nice to see the spatial sensitivity limit of the transducer with respect to the various distance from the defect. In other words, at what distance (minimum) from the defect should the transducer be located to correctly recognize the defect?
Response: The comment is considered. Actually, one of the purposes of this test is to estimate the effective distance of PZT patch to debonding damage, however the corresponding test results were not given in the manuscript, and now it is added in the revised manuscript.
Reviewer 4 Report
It was a pleasure to review the manuscript, which is well written and presents relevant results for the field of SHM on the basis of EMI techique. In my opinition, I recommend the publication in present form after insertion of minor revisions: a complete review upon the definition of some acronyms (EMI, CFRP, PZT...just to name a few) should be performed by the authors (usually, the full definitions of these acronyms are not started in first capital letter).
Author Response
Response to Reviewer 4’s Comments
The authors greatly appreciate the time and effort by the reviewer for improving the quality of this manuscript. All the comments indicated in the reviewers’ report have been properly addressed and revised in the latest manuscript. The revisions are printed with Red Color.
Comment: a complete review upon the definition of some acronyms (EMI, CFRP, PZT...just to name a few) should be performed by the authors (usually, the full definitions of these acronyms are not started in first capital letter).
Response: The authors greatly appreciate the reviewer’s comment. We have proofread the manuscript carefully, and the writing has been improved.
Round 2
Reviewer 2 Report
ok
Reviewer 3 Report
It is OK now.